# CogMol: Target-Specific and Selective Drug Design for COVID-19 Using Deep Generative Models

Vijil Chenthamarakshan[†], Payel Das[†], Samuel C. Hoffman, Hendrik Strobelt[*], Inkit Padhi, Kar Wai Lim[§], Benjamin Hoover[*], Matteo Manica[‡], Jannis Born[‡], Teodoro Laino[‡], Aleksandra Mojsilovic

IBM Research, Yorktown Heights, New York; [§]IBM Research, Singapore
[*]IBM Research, MIT-IBM Watson AI Lab, Cambridge; [‡]IBM Research Europe
{ecvijil,daspa,aleksand}@us.ibm.com, {shoffman,hendrik.strobelt}@ibm.com,
{inkpad,kar.wai.lim,benjamin.hoover}@ibm.com, {tte,jab,teo}@zurich.ibm.com

## Abstract

The novel nature of SARS-CoV-2 calls for the development of efficient de novo drug design approaches. In this study, we propose an end-to-end framework, named CogMol (Controlled Generation of Molecules), for designing new drug-like small molecules targeting novel viral proteins with high affinity and off-target selectivity. CogMol combines adaptive pre-training of a molecular SMILES Variational Autoencoder (VAE) and an efficient multi-attribute controlled sampling scheme that uses guidance from attribute predictors trained on latent features. To generate novel and optimal drug-like molecules for unseen viral targets, CogMol leverages a protein-molecule binding affinity predictor that is trained using SMILES VAE embeddings and protein sequence embeddings learned unsupervised from a large corpus. We applied the CogMol framework to three SARS-CoV-2 target proteins: main protease, receptor-binding domain of the spike protein, and non-structural protein 9 replicase. The generated candidates are novel at both the molecular and chemical scaffold levels when compared to the training data. CogMol also includes *in silico* screening for assessing toxicity of parent molecules and their metabolites with a multi-task toxicity classifier, synthetic feasibility with a chemical retrosynthesis predictor, and target structure binding with docking simulations. Docking reveals favorable binding of generated molecules to the target protein structure, where 87–95% of high affinity molecules showed docking free energy < -6 kcal/mol. When compared to approved drugs, the majority of designed compounds show low predicted parent molecule and metabolite toxicity and high predicted synthetic feasibility. In summary, CogMol can handle multi-constraint design of synthesizable, low-toxic, drug-like molecules with high target specificity and selectivity, even to novel protein target sequences, and does not need target-dependent fine-tuning of the framework or target structure information.

## 1 Introduction

Generating novel drug molecules is a daunting task that aims to create new molecules (or optimize known molecules) with multiple desirable properties that often compete and tightly interact with each other. For example, optimal drug molecules should have binding affinity to the target protein of interest (target specificity), low binding affinity to other targets (off-target selectivity), be easy to synthesize, and also exhibit high drug likeliness (QED). This makes drug discovery a costly (2-3 billion USD) and time-consuming process (more than a decade) with a low success rate (< 10%) [1].

---

[†]Equal Contribution

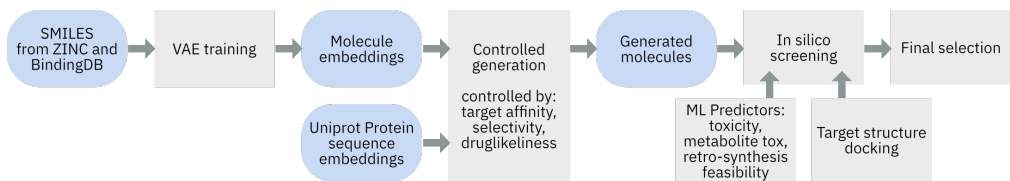

Figure 1: Workflow of the drug candidate generation and screening pipeline

Traditional *in silico* molecule design and screening rely on rational design methods that need physics-based simulations, heuristic search algorithms, and considerable domain knowledge. However, optimizing over the discrete, unstructured and sparse molecular space remains an intrinsically difficult challenge. Therefore, there is much interest in developing automated machine learning techniques to efficiently discover sizeable numbers of plausible, diverse and novel candidate molecules in the vast ($10^{23}$-$10^{60}$) space of molecules [2]. Bayesian optimization, Reinforcement Learning, and gradient-based optimization methods have been proposed for automating drug molecule design with desired properties (e.g., high drug-likeliness, synthetic accessibility, or solubility) [3–5]. These methods either optimize directly on the high-dimensional input space or on the low dimensional representation learned using a latent variable model such as a probabilistic autoencoder.

One crucial aspect of designing drug candidates is to account for the right context, e.g., protein, gene, metabolic or disease pathway information. For example, in target protein-specific drug design, the goal is to generate molecules with high binding affinity to a specific target protein. This requires fine-tuning a generative model on a small library of ligands to enable target-specific sampling. For novel or unrelated proteins, such as the SARS-CoV-2 viral proteins involved in the recent COVID-19 pandemic, binding affinity data is unavailable. At the same time, these novel target proteins are not related to the proteins in existing binding affinity databases. Thus, handling novel targets in the current generative frameworks becomes non-trivial.

Designing drug candidates for novel targets gets even more challenging, as the drug designed for the novel target can bind to other undesired targets. Small molecule drugs have been shown to bind on average to a minimum of 6-11 distinct targets in addition to their intended target [6]. This molecular "promiscuity" of drugs causes unintended therapeutic effects or multiple drug–target interactions leading to off-target toxicities and decreased effectiveness [7, 8]. Accounting for this important aspect of off-target selectivity becomes non-trivial if the generative model is trained only on a small ligand library optimized for a single target or only on good binder molecules for a limited set of targets.

## 2 CogMol - Molecule Generation Pipeline

To address the challenges stated above, we propose an alternative method named **Co**ntrolled **Ge**neration of **Mol**ecules (CogMol) for designing small molecule drugs in a context, *i.e.* target protein in this study, -dependent manner. CogMol accounts for both target specificity and selectivity, even for novel or low-coverage target sequences. As depicted in Figure 1, CogMol includes the following components:

**1.** A Variational Autoencoder (VAE), first trained unsupervised and then jointly with a set of attribute regressors (QED and Synthetic Accessibility, SA), that learns a disentangled latent space of the molecules using the SMILES representation.
**2.** A protein-molecule binding affinity regressor trained on the VAE latent features of molecules and protein sequence embeddings trained on a large unlabeled corpus, which is used for estimating target specificity and off-target selectivity.
**3.** An efficient sampling scheme to generate molecules with desired attributes from the model of the VAE latent space, using guidance from a set of attribute (affinity, selectivity, QED) predictors.

Instead of training the binding affinity regressor on the sequence embeddings of a few thousand target proteins reported in the binding affinity database, CogMol uses pre-trained protein sequence embeddings, [9] learned on an unlabeled corpus of 24M Uniprot protein sequences from UniRef50 database, to train the affinity predictor. Since these pre-trained protein embeddings are capable of better capturing sequence, structural, and functional relationships [9, 10], using them in CogMol allows targeted generation of molecules even for new/unseen targets and does not require model retraining for every individual target. Finally, CogMol proposes an efficient way of modeling off-

target selectivity and using this as a control for targeted generation, leveraging the same trained protein-ligand binding affinity predictor.

CogMol is also empowered with an *in silico* screening protocol for generated molecules, which involves: (**i**) toxicity prediction on several *in vitro* and clinical endpoints for parent molecules and their predicted metabolites using a multi-task deep learning-based classifier, (**ii**) synthetic feasibility prediction using a chemical retrosynthesis predictor; and (**iii**) blind docking simulations to estimate binding of the generated molecules to the target protein structure. We hope that accounting for multiple important properties that play a role in the efficacy or viability of a drug such as target affinity, off-target selectivity, toxicity of parent molecules and their metabolites, and synthetic practicality, within an AI framework will help the *in silico* drug design process to be faster and less costly, leading to shorter discovery pipelines with high success rate.

**CogMol for COVID-19 Antiviral Molecule Design.** Given the urgency with the ongoing COVID-19 pandemic, we apply CogMol to generate candidate molecules that bind to three relevant target proteins of the novel SARS-CoV-2 virus, namely NSP9 Replicase (NSP9), Main Protease ($M^{pro}$), and the Receptor-Binding Domain (RBD) of the SARS-CoV-2 S protein, with high affinity. Note, these targets are not present in the binding affinity database, and both NSP9 and RBD are more novel than $M^{pro}$ (See Supp. Mat. A). We also used CogMol to generate molecules for human Histone deacetylase 1 (HDAC1) protein implicated in cancer, for which the number of molecules with desired criteria in the training database is very low.

Our contributions in this work are: (**i**) An end-to-end framework for drug-like small molecule design that accounts for multiple relevant and critical factors such as target affinity, off-target selectivity, toxicity of the parent molecules and their metabolites across different endpoints, target structure binding, and synthetic practicality. (**ii**) This is, to our knowledge, the first deep generative approach that generates novel, specific, and selective drug-like small molecules for a *unseen* target sequence without model retraining. (**iii**) A system capable of generating drug-like molecules with high target affinity and selectivity for selected targets that are either relatively novel or have a low ligand coverage. (**iv**) Although our framework learns from 1D protein sequence information only, generated molecules are still capable of binding to the druggable binding pockets of the 3D target structure with favorable binding free energy (BFE). (**v**) For three novel and very relevant COVID-19 targets (as well as for a cancer target with low coverage of optimal ligands), we are able to identify a set of generated novel and optimal drug-like molecules with high predicted target affinity and selectivity, that docks favorably to the target structure, is synthetically practical and has low predicted parent and metabolite toxicity with respect to FDA-approved drugs.

## 3   Related Work

Earlier approaches to generate molecules involved recurrent neural networks (RNN) [11, 12], whereas recent works employ deep generative frameworks, such as the Variational Autoencoder (VAE) [3, 13–15] and the Generative Adversarial Network (GAN) [16, 17]. Most of those works employ SMILES representation. Generating syntactically valid molecules under SMILES grammar is challenging and there have been attempts to ensure validity via constraints [18, 19]. Recently, there has been increasing interest in molecular graph-based generative methods [20–25]. Unfortunately, graph-based models are not permutation-invariant of their node labels, the training has a quadratic complexity concerning the number of nodes, and generating semantically valid graphs is challenging. [26] is considered as a state-of-the-art architecture in this context, which represents a molecular graph as fragments connected in a tree structure.

For targeted generation of molecules with a set of desired attributes, Reinforcement Learning (RL) and Bayesian Optimization (BO) methods have often been employed on top of a SMILES- or Graph-based molecule generator [27–30, 4, 3, 31, 5], but typically incur high computational cost. Semi-supervised learning has also been used [15, 14, 21], which involves optimizing complicated loss objectives. CogMol is instead inspired by the Controlled Latent attribute Space Sampling (CLaSS) method [32]. Our proposed methodology aims at computationally efficient targeted generation with multiple constraints from the latent space, requiring minimal model training and no supervision.

To generate drugs specific to a particular target, generative models in existing works [4, 21, 33] are typically fine-tuned on the subset of molecules that bind to that specific target sequence or structure and hence cannot generalize to unseen targets. Recently, target-specific de novo drug

design has been defined as a translation problem between amino acid "language" and molecular SMILES [34], where the latent code $z$ of a protein is considered as a "context" to generate a binding ligand. However, the protein embeddings are learned only from the ∼1100 human protein sequences captured in BindingDB, which limits the model's generalization capabilities. In contrast, CogMol uses protein embeddings from Unirep trained on an unsupervised corpus of ∼24 million UniRef50 amino acid sequences [9]. This approach has been demonstrated to improve performance in downstream prediction tasks [10, 9] as well as in generative modeling [32].

## 4  Model and Methods

**Data.** We used the Moses benchmarking dataset [35] for the unsupervised VAE training, which include 1.6M molecules in the training set and 176K molecules in the test and scaffold test sets respectively from the ZINC database [36].

For target-specific compound design, we used a curated IC50-labeled compound-protein binding data from BindingDB [37], as reported in DeepAffinity [38]. The DeepAffinity models use a separate held-out set with four different protein classes to test the generalizability of their predictor. Since our objective is to build the best binding affinity (pIC50) regression model using available data, we also added the four excluded classes into our training data.

**Variational Autoencoder for Molecule Generation.** A Variational Autoencoder (VAE) [39] frames an autoencoder in a probabilistic formalism that constrains the expressivity of the latent space, $z$. Each sample defines an encoding distribution $q_\phi(z|x)$ and for each sample, this encoder distribution is constrained to be close to a simple prior distribution $p(z)$. We consider the case of the encoder specifying a diagonal Gaussian distribution only, i.e. $q_\phi(z|x) = N(z; \mu(x), \Sigma(x))$ with $\Sigma(x) = \text{diag}[\exp(\log(\sigma_d^2)(x)))]$. The encoder neural network produces the log variances $\log(\sigma_d^2)(x)$. The marginal posterior is $q_\phi(\mathbf{z}) = \frac{1}{N}\sum_{j=1}^{N} q_\phi(\mathbf{z}|\mathbf{x}_j)$. The standard VAE objective is defined as follows (where $D_{\text{KL}}$ is the Kullback-Leibler divergence), $\mathcal{L}_{\text{VAE}}(\theta, \phi) = \mathbb{E}_{p(x)}\left\{\mathbb{E}_{q_\phi(z|x)}[\log p_\theta(x|z)] - D_{\text{KL}}(q_\phi(z|x)||p(z))\right\}$.

We also jointly trained two property predictors, one for QED and one for SA, each parameterized by a feed-forward network, along with the VAE, to predict $y(x)$ from the latent embedding of $x$. As shown in Supp. Mat. K, the BindingDB molecules have a different distribution of QED when compared to molecules in ZINC. To better reflect this diversity in the latent embeddings of the VAE, we continued training of the VAE model with QED and SA predictors on BindingDB molecules. We report the architecture and performance of the final VAE model in Supp. Mat. B.

**Attribute Predictors.** We train multiple property predictors for controlling generation. The architecture and performance of these predictors are reported in Supp. Mat. F. First, to test the information content of the VAE latent space, we trained multiple attribute (QED, logP, and SA) predictors on the latent embeddings. These models show low root-mean-square-error (RMSE) on test data for all three attribute predictors.

Next, we trained a binding affinity regression model using the pIC50 ($= -log(IC50)$) data from BindingDB. This model takes a representation of a target protein sequence and latent embedding, $\mathbf{z}$, of a molecule as input, and predicts the binding affinity between the protein-molecule pair. We used pre-trained protein embeddings from [9] to initialize the weights for proteins. This model, along with the model for QED is used in the controlled generation pipeline. We also trained a binding affinity predictor using SMILES ($\mathbf{x}$) instead of latent ($\mathbf{z}$) embedding as the input molecular representation. This model was used during the *in silico* screening process as it has a higher accuracy than the model trained on the latent embeddings, comparable to a recent model described in [38].

**Selectivity Modeling.** Selectivity to a particular target is often modeled only in the later stages of a drug development pipeline. It has been suggested that improvement in the early accounting of off-target interactions represents an opportunity to reduce safety-related attrition rates during pre-clinical and clinical development [8]. Given the novel nature of COVID-19, it is even more important to account for off-target selectivity in the early design state in order to minimize undesired interactions with host targets. Thus, we believe that incorporating selectivity during the candidate generation stage will contribute to a reduction in the failure rate of drug candidates. We define selectivity as the excess binding affinity (BA) of a molecule ($m$) to a target of interest ($T$) over its average binding affinities to a random selection of $k$ targets [40]: $Sel_{T,m} = BA(T, m) - \frac{1}{k}\sum_{i=1}^{k} BA(T_i, m)$.

**Controlled Generation.** Our objective is to generate molecules that simultaneously satisfy multiple (often conflicting) objectives. Specifically, we want generated molecules controlled by high binding affinity to a selected novel SARS-CoV-2 target, high drug-likeness, and high off-target selectivity.

For this purpose, we performed conditional generation using Conditional Latent (attribute) Space Sampling — CLaSS proposed recently in [32]. In short, CLaSS leverages the attribute predictors trained on the latent features and uses a rejection sampling scheme to generate samples with desired attributes from a density model of the latent space. Since the goal is to sample conditionally $p(\mathbf{x}|\mathbf{a})$, where $\mathbf{a} \in \mathbb{R}^n = [a_1, a_2, \ldots, a_n]$, a set of independent attributes, CLaSS approaches this task through conditional sampling in latent space: $p(\mathbf{x}|\mathbf{a}) = \mathbb{E}_{\mathbf{z}}[p(\mathbf{z}|\mathbf{a})p(\mathbf{x}|\mathbf{z})] \approx \mathbb{E}_{\mathbf{z}}[\hat{p}_\xi(\mathbf{z}|\mathbf{a})p_\theta(\mathbf{x}|\mathbf{z})]$. Where $\hat{p}_\xi(\mathbf{z}|\mathbf{a})$ uses rejection sampling from parametric approximations to $p(\mathbf{z}|\mathbf{a})$. The term $\hat{p}_\xi(\mathbf{z}|\mathbf{a})$ is approximated using a density model $Q_\xi(\mathbf{z})$, such as a Gaussian mixture model and per-attribute classifier model $q_\xi(a_i|\mathbf{z})$. This is approached by using Bayes' rule and then conditional independence of the attributes [32]. Rejection sampling is then performed through the proposal distribution: $g(\mathbf{z}) = Q_\xi(\mathbf{z})$ that can be directly sampled. Since we impose multiple attribute constraints for sampling, intuitively, the acceptance probability is equal to the product of the attribute predictors' scores, while sampling from explicit density $Q_\xi(z)$. As long as there is a region in $\mathbf{z}$ space where $Q_\xi(\mathbf{z}) > 0$ and probabilities from all predictors are $> 0$, samples will be accepted in this scheme. Consequently, CLaSS can sample from the targeted region of the autoencoder latent space, which was trained unsupervised. Learning to control for one or more attribute(s) in CLaSS is computationally efficient, as it does not require a surrogate model or policy learning and neither adds complicated loss terms to the original objective.

***In Silico* Screening.** Molecular toxicity or side effect testing is conventionally carried out via different endpoint experiments, *e.g., in vitro* molecular assays, *in vivo* animal testing, clinical trials, and adverse effect reports. However, these experiments are costly and time-consuming. We instead used a multitask deep neural network (MT-DNN) for binary (yes/no) toxicity prediction as an early screening tool to prioritize the testing of molecules that are less likely to be harmful and to speed up the process of finding a COVID-19 therapeutic (For details see Supp. Mat. G). A multitask model is expected to improve the prediction by exploiting the correlation between different endpoints. The MT-DNN was used to predict the toxicity of 12 *in vitro* endpoints from the Tox21 challenge [41]. We also predicted whether the generated molecules would fail clinical trials, using the ClinTox data [42].

The generated molecules were screened further for target affinity and selectivity using the $\mathbf{x}$-level binding affinity predictor (See Supp. Mat. F). To investigate the possible binding modes of the generated molecules with the target protein structure, we performed 5 independent runs of blind docking of the generated achiral molecules with the target structure using Autodock Vina [43]. To evaluate the synthetic accessibility, the generated molecules were analyzed using a retrosynthetic algorithm [44] based on the Molecular Transformer [45] trained on patent chemical reaction data.

## 5 Results and Discussion

### 5.1 Benchmarking Molecular VAE Model

The architecture and performance metrics of the final VAE model that is adaptively pre-trained from ZINC to BindingDB with SA and QED supervision are provided in Supp. Mat. B, along with a comparison to a number of baseline models. The majority of the generated molecules are chemically valid (90%), unique (99%), pass relevant filters (95%), and show a slightly higher diversity (Table B.1). One interesting observation from Table B.2 is that the generated molecular ensemble has a higher Fréchet ChemNet Distance (FCD) [46] with respect to chemical scaffolds present in both ZINC and BindingDB training molecules. This implies that adaptive pre-training from ZINC to BindingDB enables the discovery of novel chemical scaffolds, which is further confirmed by comparing the Tanimoto Similarity between generated scaffolds and reference scaffolds (Supp. Mat. Figure B.1). A few synthetically plausible and novel scaffolds from the generated set are shown in Supp. Mat. Figure B.2

### 5.2 Attributes of COVID-Targeted Molecules

**CogMol-Controlled Attributes — Target Affinity, Selectivity, and QED.** Table 1 reports higher proportion of molecules with desired attributes in the set accepted in CLaSS, when compared to a

Table 1: Normalized fraction of molecules that are accepted in CLaSS with different set of controls (Affinity, QED, and Selectivity). The values of controls are normalized between 0 and 1. As we increase the extent of controls, a relatively higher proportion of molecules meeting all criteria are in the accepted set compared to a randomly sampled set.

|  | Aff >0.5 | | Aff >0.5 & QED >0.8 | | Aff >0.5 & QED >0.8 & Sel >0.5 | |
|---|---|---|---|---|---|---|
|  | Accepted | Random | Accepted | Random | Accepted | Random |
| NSP9 | 0.567 | 0.355 | 0.45 | 0.211 | 0.069 | 0.007 |
| RBD | 0.546 | 0.369 | 0.429 | 0.217 | 0.09 | 0.009 |
| M$^{pro}$ | 0.603 | 0.366 | 0.472 | 0.216 | 0.104 | 0.011 |

Table 2: CogMol-generated SMILES found in PubChem and their predicted affinity (pIC50), lowest docking free energy (kcal/mol), PubChem Compound ID (CID), and reported biological activity.

| Target | Pred. Affinity | Docking Energy | CID | Biological Activity |
|---|---|---|---|---|
| NSP9 Dimer | 6.51 | −7.7 | 12042753 | Antagonist of rat mGluR |
|  | 7.06 | −5.6 | 44397285 | Active to human S6 kinase |
|  | 7.18 | −6.4 | 10570770 | Matrix metalloproteinase inhibitor |
| Main Protease | 7.24 | −6.1 | 10608757 | Dihydrofolate reductase inhibitor |
|  | 6.91 | −6.9 | 872399 | Shiga toxin inhibitor |
| RBD | 7.82 | −7.5 | 76332092 | Plasmepsin inhibitor |

randomly sampled set, implying that CLaSS does generate a more optimal set than random sampling from the latent space, and the success depends on the target context. We further selected around 1000 CogMol-generated molecules for each target as explained in Supp. Mat. C. The density plots in Figure 2 of the selected set indicate that generating high-affinity ligands is more challenging for NSP9 (Figure 2a), while M$^{pro}$ ligands are more selective in general (Figure 2b), which is likely due to relative novelty of the target sequences with respect to BindingDB training sequences (see Supp. Mat. A). The QED distribution also highlights target sequence dependence of the generated molecules, as the molecules targeting RBD show a peak at a lower QED value in the distribution. Several randomly chosen samples for each SARS-CoV-2 target are shown in Supp. Mat. Figure D.1.

**Novelty.** The novelty distributions, as estimated using the Tanimoto Similarity [47] between molecular fingerprints, of the generated molecules with respect to both the PubChem [48] database and our training set are shown in Supp. Mat. Figures E.1 and E.2. When compared with the training database of size $\sim$ 1.9 M, we find that the likelihood of generating molecules with a novelty value of 0 is $\leq$ 2%. With respect to the larger PubChem database consisting of $\sim$ 103 M molecules, the majority of which were not included in model training, we find the percentage of generated molecules with novelty value of 0 is 9.5%, 3.7%, and 8.3% generated molecules for M$^{pro}$, RBD, and NSP9, respectively. Higher FCD of those generated molecules with respect to test scaffolds in ZINC/BindingDB (Supp. Mat. Table E.1) further confirms presence of novel chemical scaffolds in them.

**CogMol Identifies PubChem Molecules with Potential Anti-COVID Activity.** Only 19, 5, and 15 of the generated molecules match exactly with an existing SMILES string in PubChem, for

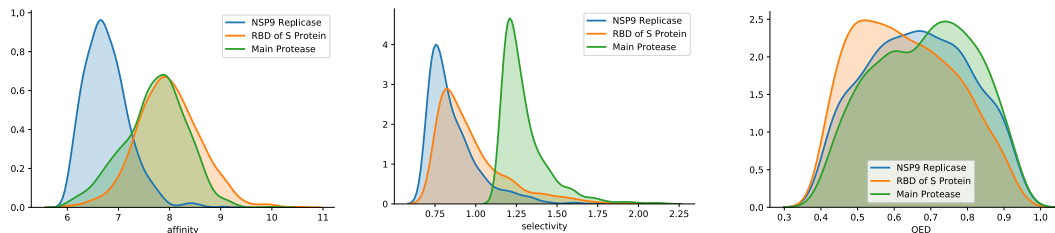

Figure 2: Density plots of (a) binding affinity, (b) off-target selectivity, (c) QED for selected molecules.

Table 3: Docking analysis: Size, average ($E$) ($\pm$ standard deviation) binding free energy (BFE), minimum BFE, fraction of generated molecules with BFE $< -6$ kcal/mol for each cluster. In parentheses after target name:% of generated molecules for the respective target that had a BFE $< -6$ kcal/mol. Only top 2 clusters are shown (see Supp. Mat. H.1).

| Target | | Size (%) | $E$ (kcal/mol) | Min (kcal/mol) | Low Energy (%) |
|---|---|---|---|---|---|
| NSP9 Dimer (87%) | cluster 0 | 67 | $-6.8 \pm 0.7$ | $-8.6$ | 88 |
| | cluster 1 | 22 | $-6.9 \pm 0.9$ | $-8.8$ | 85 |
| Main Protease (91%) | cluster 0 | 76 | $-7.2 \pm 0.8$ | $-9.5$ | 93 |
| | cluster 1 | 18 | $-6.9 \pm 0.8$ | $-9.2$ | 86 |
| RBD (95%) | cluster 0 | 30 | $-6.9 \pm 0.6$ | $-8.3$ | 93 |
| | cluster 1 | 36 | $-7.2 \pm 0.6$ | $-9.1$ | 97 |

$M^{pro}$, RBD, and NSP9, respectively. Some of these SMILES are reported with *biological activity* in PubChem, as shown in Table 2, which calls for further investigation. For example, the molecule with PubChem Compound ID (CID) 76332092 is a known Plasmepsin-2 and Plasmepsin-4 inhibitor and has also shown antimalarial activity against chloroquine-sensitive Plasmodium falciparum. As RBD from S protein binding to angiotensin-converting enzyme-2 (ACE-2) receptor is needed for the viral entry to the host cells [49], both RBD and ACE-2 receptor are being actively investigated as COVID-19 targets. Chloroquine (and its hydroxy derivative) is a known ACE-2 inhibitor and has been already considered as a promising COVID-19 drug [50]. CID 76332092 deserves further investigation in the context of SARS-CoV-2 as it shows a predicted pIC50 of 7.82 and lowest docking binding free energy (BFE) of -6.80 kcal/mol (Figure H.3) to the ACE-2 binding pocket of RBD (Table 2). The generated molecule with highest predicted affinity for RBD (with a pIC50 of 10.49 and a docking BFE of -6.9 kcal/mol in the top binding mode with ACE-2 binding pocket) also shares a strong maximum common subgraph similarity [51] with Telavancin, an approved skin infection and Pneumonia drug, as shown in Figure E.3. These results indicate that CogMol can generate promising and biologically relevant drug candidates beyond the training dataset.

**Docking with Target Structure.** Table 3 summarizes these results. In the best (lowest BFE) docking pose, 87%, 91%, and 95% of generated molecules show a minimum BFE of $< -6$ kcal/mol for NSP9 dimer, $M^{pro}$, and RBD, respectively. For each target, we classified each molecule by its binding location, fitting the geometric centers of docked molecules drawn from a larger set of 875K samples to a mixture of 4, 5, and 6 Gaussian models, respectively (see Supp. Mat. H). We also report the average and minimum BFE, as well as the fraction of generated molecules with a BFE of $< -6$ kcal/mol for each cluster (Table 3). Results show that even though CLaSS used only target sequence information for controlled generation, generated molecules do identify the relevant and known druggable binding pockets within the 3D target structure and bind to those favorably.

### 5.3 CogMol-generated Molecules Targeting Human HDAC1

Human HDAC1 plays key role in eukaryotic gene expression and is implicated in cancer. Though it is present in BindingDB, there are only a handful of molecules with high QED and high pIC50, see Table K.1. We applied CogMol to generate optimal ligands targeting HDAC1. Table K.1 shows that CogMol-generated molecules comprise a larger proportion of molecules satisfying high pIC50 and QED criteria, implying CogMol can discover novel and optimal molecules even in a low-data regime.

### 5.4 Synthesizability and Toxicity of Generated Molecules

The number of steps/reactions needed to complete the synthesis (synthetic design) provides an estimate of the complexity of the molecules respect to commercially available materials (more details and the parameters adopted, can be found in the Supp. Mat. I). In Figure 3a (legend), we report the percentage of feasibility for 4 sets of generated molecules, each targeting a different protein - NSP9, RBD, $M^{pro}$, and HDAC1. For SARS-CoV-2 targets, molecules were selected by considering the top-100 molecules based on SA. For HDAC1, calculation was done on a set of 100 generated molecules. We also estimated feasibility for a selection of FDA-approved and commercially available drugs [52] (FDA), used as a baseline. For the FDA set, the fraction of molecules predicted as feasible is $\sim 78\%$. This is expected, since most of these molecules have been protected with patents and are

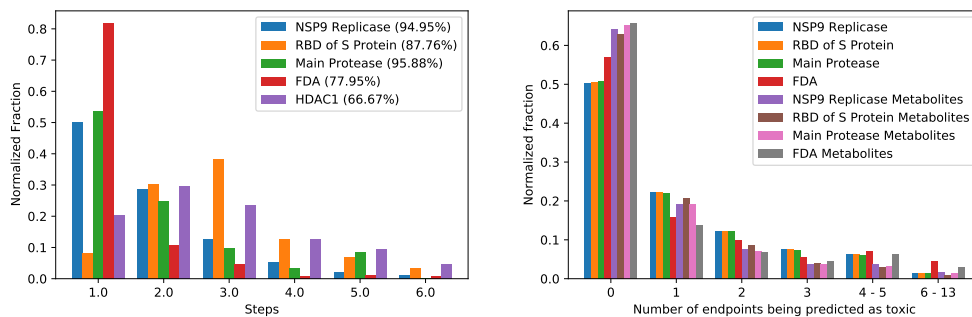

Figure 3: (a) Bar plots describe the percentage of molecules synthesizable for the exact number of retrosysthesis steps. Legend reports the fraction of molecules for each set marked as synthetically accessible. (b) Percentage of parent molecules or their metabolites as a function of number of endpoints in which predicted to be toxic.

therefore chemically accessible with the reaction knowledge available in patents. The molecules generated by CogMol for the three COVID-19 targets perform better than the FDA with successful rates > 85-90%. The HDAC1 set instead shows a value of ∼ 67%. The higher success rate for the COVID target sets indicates that the molecules in these classes are easier to synthesize from commercially available materials than the molecules belonging to the FDA class. The HDAC1 set, while still showing a relatively high synthesizability rate, demonstrates additional need of chemical knowledge uncovered in patents. The distribution of the number of steps needed for each set reveals two interesting observations, as shown in Figure 3a: FDA, M$^{pro}$ and NSP9 molecules show a peak at 1 step and HDAC1 and RBD showing a peak respectively at 2 and 3 steps. In comparison, > 80% of the successfully synthesizable FDA approved molecules can be made in a single step from commercially available molecules, likely because their precursors are also made commercially available after approval. The M$^{pro}$ and NSP9 sets are similar to the FDA approved drugs. They are characterized by a lower degree of complexity, indicating close relation to commercially available molecules, when compared to HDAC1 and RBD. Overall, the retrosynthetic analysis of the generative model outcomes clearly shows that the generated structures are chemically relevant and synthetically feasible. Additional results revealing correlations between number of synthesis steps and properties of molecules for the COVID-19 targets can be found in Supp. Mat. I.

Figure 3b shows toxicity analyses of the CogMol-generated molecules, their metabolites, as well as of FDA-approved drugs by using the MT-DNN model. A molecule was considered toxic if it was predicted to be toxic in ≥ 2 endpoints. The metabolites were predicted by using a recently proposed work that models the human metabolite prediction task of small molecules as a sequence translation problem and uses a Seq2Seq Transformer model[‡] originally pre-trained on chemical reaction data and further fine-tuned on metabolite reaction data to predict the outcome of human metabolic reactions [53]. Results in Figure 3b show that majority (∼ 70%) of the generated molecules, as well as their predicted metabolites (∼ 80% of them) are predicted toxic only in 0-1 endpoints out of a total of 13, which is comparable to the FDA-approved drugs.

### 5.5 Sharing and Visualization of Generated Molecules

We share around ∼ 3500 generated molecules under an open license for the research community to download and evaluate. In order to help domain experts, we also created a publicly available Molecule Explorer tool to facilitate screening and filtering of the molecules, perform novelty analysis, and identify closest molecules in PubChem. A screen cast of the Molecule Explorer tool is provided.[§]

## 6   Conclusions and Future Work

In this paper, we proposed CogMol, a framework for Controlled Generation of Molecules with a set of desired attributes. Our framework can handle targeted and novel compound generation for

---

[‡]Source code available at https://github.com/KavrakiLab/MetaTrans
[§]https://www.youtube.com/watch?v=cYb8_catBpI

multiple proteins using the same trained model, can generalize to unseen viral proteins, and accounts explicitly for off-target selectivity. Additionally, we provide an *in silico* screening method that accounts for target structure binding, *in vitro* and clinical toxicity of parent molecules and their metabolites, and synthesis feasibility. When applied to COVID-19 novel viral protein sequences, CogMol generated novel molecules were able to bind favorably to the relevant druggable pockets of the target structure. The generated compounds are also comparable to FDA-approved drugs in parent molecule, metabolite toxicity, and synthetic feasibility. In summary, our framework provides an efficient and viable computational framework for de novo multi-objective design and filtering of optimal drug compounds that are specific and selective to novel/unseen targets. Future work will address accounting for additional contexts (on top of target protein), adding other pharmacologically relevant controls, and also weigh those according to their relative importance to make CogMol framework more efficient in term of generating promising drug candidates.

# 7    Statement of Broader Impact

We discuss the broader impact of our work from the following perspectives.

**Benefits** To date, SARS-CoV-2 has infected millions and killed hundreds of thousands around the globe and continues to cause a severe economical crisis [54]. are still undergoing investigation [55]. Therefore, it is timely to explore for efficient *de novo* drug design approaches to combat COVID-19 and future pandemics. The CogMol framework is adapative, generic, and could pave the road for accelerated discovery of new antivirals optimized against specific SARS-CoV-2 (or other novel virus) targets. This could have a major impact on our global effort against COVID-19 and future novel pandemics and save human lives.

We demonstrated that our framework can generate target-specific and selective compounds for *unseen* protein targets, a novel property that may be key for swift reactions to possible SARS-CoV-2 mutants. We further provide *early assessment* of novel AI-generated compounds on target structure binding, and synthetic feasibility and toxicity in the context of FDA-approved drugs, in order to identify a list of promising compounds that is of reasonable size and can be immediately sent to wet lab for synthesis and validation. We showed the efficiency of the framework in terms of handling multiple constraints at once and can be easily extended to adding more controls to account for additional factors considered crucial in drug discovery such as ADME properties. Thus, our approach systematically bridges biology and machine learning to accelerate drug discovery.

We further share with the community a list of CogMol-generated compounds (and their attributes) designed for three novel SARS-Cov-2 targets, as well as a molecular explorer tool to visualize, experience, and provide feedback on these molecules. This sets our vision for an open community of discovery that facilitates interactions between AI researchers and medicinal scientists.

**Risks and the Potential to Cause Harm** While our approach offers enormous potential to speed up the development of new drugs, it must be realized that drug candidate generation and *in silico* screening are merely first steps in the development of viable therapeutics. No wetlab evaluation of the generated molecules have been done. The ability of the public to order these novel compounds online, poses a risk that it might be tried by people who are not sufficiently educated about the dangers of exposing themselves to these molecules in an uncontrolled setting. The public must be educated to not to treat these candidates as approved drugs or miracle cures. Further, since our framework allows generation of molecules satisfying arbitrary objectives, this capability can be misused by bad actors to design potentially harmful chemicals.

**Consequences of Failure** It is possible that our framework will not be able to generate molecules with desired properties either because of bias in training data or because of the inaccuracy of the predictors used for controlled generation. Also, there could be a divergence between the machine learning predicted properties and wet lab experimental evaluations. One way to address this problem is to cross check these properties using multiple independent machine learning or other models.

# Acknowledgments and Disclosure of Funding

We thank Eleni E. Litsa and Lydia E. Kavraki for helping to predict the metabolites of CogMol-designed molecules. This work was funded internally by IBM Research.

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
