[Supplementary Material]

## Supplementary Materials

## A   Protein Targets Chosen for Generation

Figure A.1 shows the amino acid sequences corresponding to the three SARS-CoV-2 targets.

We also computed the similarity of these targets with respect to our training data from BindingDB using NCBI-BLAST tool[¶]. The BLAST tools computes Expect value (E-value) - a measure of statistical significance of the match between the query sequence and database sequences. Larger E-value indicates a higher chance that the similarity between the hit (from the database) and the query is merely a coincidence, i.e. the query is not homologous or related to the hit. The lowest Expect value with respect to BindingDB protein sequences using the default parameters of BLAST are: $M^{pro}$ = 0.51 (query coverage = 40%), RBD = 1.9 (query coverage = 26%), NSP9 = 3.2 (query coverage = 10%).

```
> SARS-CoV-2 Main Protease
SGFRKMAFPSGKVEGCMVQVTCGTTTLNGLWLDDVVYCPRHVICTSEDMLNPNYEDLLIRKSNHNFLVQAGNVQLRVIGHSMQNCVLKLK
VDTANPKTPKYKFVRIQPGQTFSVLACYNGSPSGVYQCAMRPNFTIKGSFLNGSCGSVGFNIDYDCVSFCYMHHMELPTGVHAGTDLEGN
FYGPFVDRQTAQAAGTDTTITVNVLAWLYAAVINGDRWFLNRFTTTLNDFNLVAMKYNYEPLTQDHVDILGPLSAQTGIAVLDMCASLKE
LLQNGMNGRTILGSALLEDEFTPFDVVRQCSGVTFQ

> SARS-COV-2 NSP9 Replicase
SNAMNNELSPVALRQMSCAAGTTQTACTDDNALAYYNTTKGGRFVLALLSDLQDLKWARFPKSDGTGTIYTELEPPCRFVTDTPKGPKVK
YLYFIKGLNNLNRGMVLGSLAATVRLQ

> Receptor-Binding Domain (RBD) of SARS-COV-2 S protein
RVVPSGDVVRFPNITNLCPFGEVFNATKFPSVYAWERKKISNCVADYSVLYNSTFFSTFKCYGVSATKLNDLCFSNVYADSFVVKGDDVR
QIAPGQTGVIADYNYKLPDDFMGCVLAWNTRNIDATSTGNYNYKYRLFRKSNLKPFERDISTEIYQAGSTPCNGVEGFNCYFPLQSYGFQ
PTNGVGYQPYRVVVLSFELLNAPATVCGPKLSTDLIK
```

Figure A.1: Sequences of SARS-CoV-2 targets

## B   Variational Autoencoder

We used a bidirectional Gated Recurrent Unit (GRU) with a linear output layer as an encoder. The decoder is a 3-layer GRU RNN of 512 hidden dimensions with intermediate dropout layers with dropout probability 0.2. The performance characteristics of the VAE model with respect to various metrics are given below.

Table B.1: The performance metrics of the generative model: Fraction of valid molecules, Fraction of unique molecules from a sample of 1,000 and 10,000 molecules, Internal Diversity (IntDiv1 and IntDiv2), Fraction of molecules passing filters (MCF, PAINS, ring sizes, charges, atom types)

| Model | Valid | Unique@1k | Unique@10k | IntDiv1 | IntDiv2 | Filters |
|---|---|---|---|---|---|---|
| ZINC (Unsupervised) | 0.9553 | 1.0 | 0.9996 | 0.8568 | 0.8510 | 0.9889 |
| ZINC (Supervised) | 0.95 | 1.0 | 0.999 | 0.8578 | 0.8521 | 0.9888 |
| BindingDB (Supervised) | 0.904 | 1.0 | 0.9993 | 0.8717 | 0.8665 | 0.9482 |
| CharRNN | 0.809 | 1.0 | 1.0 | 0.855 | 0.849 | 0.975 |
| AAE | 0.997 | 1.0 | 0.995 | 0.857 | 0.85 | 0.997 |
| VAE | 0.969 | 1.0 | 0.999 | 0.856 | 0.851 | 0.996 |
| JT-VAE | 1.0 | 1.0 | 0.999 | 0.851 | 0.845 | 0.978 |
| Training | 1.0 | 1.0 | 1.0 | 0.857 | 0.851 | 1.0 |

---

[¶]https://ftp.ncbi.nlm.nih.gov/blast/executables/blast+/LATEST/

Table B.2: Performance evaluation of the generative model using scaffold split metrics: Fréchet ChemNet Distance (FCD), Similarity to the nearest neighbour (SNN), Fragment similarity (Frag), Scaffold similarity (Scaff). The model trained on BindingDB is evaluated on both BindingDB and ZINC Scaffolds.

| Model | FCD | | SNN | | Frag | | Scaff | |
|---|---|---|---|---|---|---|---|---|
| | Test | TestSF | Test | TestSF | Test | TestSF | Test | TestSF |
| ZINC(Unsupervised) | 0.166 | 0.603 | 0.560 | 0.533 | 0.999 | 0.997 | 0.905 | 0.128 |
| ZINC(Supervised) | 0.2051 | 0.6222 | 0.5526 | 0.5267 | 0.999 | 0.998 | 0.8907 | 0.1319 |
| BindingDB (BindingDB Scaff.) | 0.7322 | 9.535 | 0.4335 | 0.3732 | 0.998 | 0.8493 | 0.5382 | 0.0764 |
| BindingDB (ZINC Scaff.) | 7.3416 | 7.7179 | 0.4089 | 0.4002 | 0.9593 | 0.9576 | 0.3196 | 0.0869 |
| CharRNN | 0.355 | 0.899 | 0.536 | 0.514 | 0.999 | 0.996 | 0.882 | 0.14 |
| AAE | 0.395 | 1.0 | 0.62 | 0.575 | 0.995 | 0.994 | 0.866 | 0.1 |
| VAE | 0.084 | 0.541 | 0.623 | 0.677 | 1.0 | 0.998 | 0.993 | 0.062 |
| JT-VAE | 0.422 | 0.996 | 0.556 | 0.527 | 0.996 | 0.995 | 0.892 | 0.1 |
| Training | 0.008 | 0.476 | 0.642 | 0.586 | 1.0 | 0.999 | 0.991 | 0.0 |

Figure B.1: The novelty of the scaffold of each generated molecule compared to the most similar scaffold in the training set. 26k molecules were generated and their scaffolds compared to the scaffolds of every molecule in the Zinc and BindingDB datasets. The results indicate that while many molecules have scaffolds which are present in the dataset (indicated by the spike at novelty = 0), there are many molecules that contain scaffolds not at all present in the training data.

Figure B.2: Comparing example scaffolds in the 26k generated molecules to the scaffolds within the training data. The most similar scaffold in the training dataset, as calculated by the Tanimoto Similarity of the fingerprints, is shown next to the scaffold of each generated molecule. These novel scaffolds were determined by a synthetic organic chemist to be synthetically plausible.

## C Selection of Molecules for Further Analysis

We selected around 1000 molecules for each target based on binding affinity ($pIC50 > 6$), QED ($> 0.4$), Synthetic Accessibility ($< 5$), number of toxic endpoints ($< 2$), logP ($< 5$) and Mol. Wt ($< 500$) for further analysis. The off-target selectivity was chosen to be higher than $1.15$, $0.75$ and $0.7$ for Main Protease, RBD of S Protein and NSP9 Replicase respectively.

## D Random Examples of Generated Molecules

We show a representative set of molecules generated for each target in Figure D.1

Figure D.1: Representative molecules generated for (top to bottom): NSP9 Replicase, Receptor-Binding Domain (RBD) of S protein, and Main Protease of SARS-CoV-2

## E Novelty Analysis

To assess the novelty of generated molecules, we assigned to each molecule $m$ a score $Nov_m$ representing its minimal distance (maximal similarity) to all registered compounds $p$ in the training database or another reference database of known compounds $P$: $Nov_m = 1 - \max_{p \in P} \{\text{sim}(k_m, k_p)\}$. MACCS keys [56], represented as 166 length bit vectors, were used as structural fingerprints to determine the similarity for each pair of molecule ($k_m$) and compound ($k_p$). The Tanimoto [47] coefficient between two fingerprints expresses the similarity: $\text{sim}(k_x, k_y) = \frac{|k_x \cap k_y|}{|k_x \cup k_y|}$. Note that a novelty of 0 means that the molecule's fingerprint matches exactly the fingerprint of a compound in the reference database; however, the final structure of the generated molecule can still be different.

The distribution of novelty scores for each of the targets with respect to the training set and a larger set of molecules from PubChem is given in Figures E.1 and E.2.

We further compute the FCD of the generated molecules (and scaffolds) for each target with respect to the ZINC and BindingDB datasets (See Table E.1. We note that novel scaffolds emerge in the generated molecules with respect to both ZINC and BindingDB.

It is also interesting to note that the CogMol generated molecule with the highest binding affinity to RBD has maximum subgraph similarity to a commercially available drug Telavancin (See Figure E.3).

Figure E.1: Novelty of the generated molecules for each target relative to the molecules in the training set, confirming that the model is indeed creating new molecules with novel fingerprints.

Figure E.2: Novelty relative to PubChem Dataset. Tends to be lower than novelty compared to all the training molecules, likely because there are significantly more molecules recorded in PubChem than the training subset.

Telavancin is a semi-synthetic derivative of vancomycin. It is used to treat complicated skin and skin structure infections, and hospital-acquired and ventilator-associated bacterial pneumonia caused by Staphylococcus aureus.

# F    Property Predictors

Property predictors for QED, logP and SA were trained on the latent embeddings of the VAE. These regression models have 4 hidden layers with 50 units each and ReLU nonlinearity. We further train a binding affinity predictor using the latent embeddings of the VAE and pretrained protein embeddings [9]. The protein embeddings and the molecular embeddings are concatenated and passed

Table E.1: FCD of the generated molecules for each target

| Target | FCD/Test (ZINC) | FCD/Test (BindingDB) | FCD/TestSF (ZINC) | FCD/TestSF(BindingDB) |
|---|---|---|---|---|
| NSP9 | 7.106 | 1.007 | 7.5 | 9.25 |
| RBD | 7.072 | 1.004 | 7.472 | 9.202 |
| MPro | 7.107 | 0.995 | 7.523 | 9.278 |
| HDAC1 | 8.028 | 1.924 | 8.47 | 10.167 |

Figure E.3: Maximum Common Subgraph Similarity of the CogMol-generated molecule with highest binding affinity to RBD (left) and Telavancin (right)

through a single hidden layer with 2048 hidden units and ReLU nonlinearity. In order to build a predictor for binding affinity trained directly on the smiles sequences (x), we first embed them using LSTMs. When Protein sequences are embedded using LSTM, it serve as a baseline, and we get a RMSE of 1.0104 on test data. Our best model on Binding Affinity, with RMSE of 0.8426, uses pre-trained protein embeddings from [9].

Table F.1: Performance of the attribute predictors for QED, logP, SA, and binding affinities. Binding affinity (z) is trained on the latent space of the VAE, while binding affinity (x) is trained on the actual SMILES sequences.

| Attribute | RMSE |
|---|---|
| Binding Affinity (x) Baseline | 1.0104 |
| QED | 0.0281 |
| logP | 0.3307 |
| SA | 0.0973 |
| Binding Affinity (z) | 1.2820 |
| Binding Affinity (x) | 0.8426 |

# G   Toxicity Prediction Model

The MT-DNN model contains a total of four hidden layers: two are shared across all toxicity endpoints and two are private for each of the endpoints. We used a dropout [57] probability of 0.5, and a ReLU activation function for all layers except for the last layers, in which the sigmoid activation was used. Morgan Fingerprints [58] were used as the input features to the model.

The ROC AUC, Accuracy (ACC), Balanced Accuracy (BAC), True Negative (TN), True Positive (TP), Precision (PR), Recall (RC), and the F1 score of the MT-DNN model on Tox21 and ClinTox test data are reported in Table G.1 in the Appendix. Although the AUC values are slightly worse than the existing work of [59, see Table S14], the precision (and thus true positive rate) achieved by the MT-DNN is much higher.[||] For comparison, we also report the results from a random forest (RF) model in Table G.2, showing that the MT-DNN significantly outperforms the RF model in terms of true positive rate, recall, and F1 score. Therefore, the MT-DNN model was used for assessing the generated molecules for toxicity.

Tables G.1 and G.2 show the performance of toxicity prediction using the MT-DNN and the random forest as the baseline. The MT-DNN significantly outperforms the RF model in terms of true positive rate, recall, and F1 score, while incurring a small penalty in ROC AUC and precision. Table G.3 displays the proportion of molecules being predicted toxic in a number of endpoints. We can see that the predicted toxicity of the generated molecules in all three targets are similar to that of the FDA approved drugs.

Table G.1: Performance on toxicity prediction using MT-DNN for all 12 Tox21 tasks and ClinTox task (CT-TOX). The reported metrics are ROC AUC, accuracy, balanced accuracy, true negative rate, true positive rate, precision, recall, and the F1 score. Refer to Table G.2 for a comparison with random forest, the MT-DNN achieves much better true positive, recall and F1 score with slight penalty on AUC.

| Task | AUC | ACC | BAC | TN | TP | PR | RC | F1 |
|---|---|---|---|---|---|---|---|---|
| NR-AR | 0.72 | 0.97 | 0.73 | 0.99 | 0.46 | 0.98 | 0.46 | 0.55 |
| NR-Aromatase | 0.78 | 0.95 | 0.65 | 0.99 | 0.31 | 0.96 | 0.31 | 0.41 |
| NR-PPAR-$\gamma$ | 0.75 | 0.97 | 0.61 | 0.99 | 0.23 | 0.96 | 0.23 | 0.31 |
| SR-HSE | 0.75 | 0.94 | 0.61 | 0.99 | 0.23 | 0.94 | 0.23 | 0.32 |
| NR-AR-LBD | 0.82 | 0.98 | 0.78 | 0.99 | 0.56 | 0.99 | 0.56 | 0.66 |
| NR-ER | 0.69 | 0.88 | 0.64 | 0.96 | 0.32 | 0.89 | 0.32 | 0.40 |
| SR-ARE | 0.78 | 0.86 | 0.67 | 0.95 | 0.39 | 0.88 | 0.39 | 0.47 |
| SR-MMP | 0.87 | 0.89 | 0.75 | 0.96 | 0.54 | 0.93 | 0.54 | 0.62 |
| NR-AhR | 0.85 | 0.91 | 0.73 | 0.96 | 0.50 | 0.93 | 0.50 | 0.55 |
| NR-ER-LBD | 0.78 | 0.96 | 0.69 | 0.99 | 0.38 | 0.97 | 0.38 | 0.48 |
| SR-ATAD5 | 0.75 | 0.96 | 0.61 | 0.99 | 0.22 | 0.97 | 0.22 | 0.32 |
| SR-p53 | 0.79 | 0.94 | 0.64 | 0.99 | 0.29 | 0.96 | 0.29 | 0.40 |
| CT-TOX | 0.79 | 0.92 | 0.61 | 0.97 | 0.25 | 0.84 | 0.25 | 0.31 |
| Average | 0.78 | 0.93 | **0.67** | 0.98 | **0.36** | 0.94 | **0.36** | **0.45** |

---

[||]The average precision from [59] over all 13 tasks is 0.45, which was obtained by running their code available through Github.

Table G.2: Performance on toxicity prediction for Random Forest on all 12 Tox21 tasks and ClinTox task (CT-TOX). The reported metrics are ROC AUC, accuracy, balanced accuracy, true negative rate, true positive rate, precision, recall, and the F1 score.

| Task | AUC | ACC | BAC | TN | TP | PR | RC | F1 |
|---|---|---|---|---|---|---|---|---|
| NR-AR | 0.78 | 0.98 | 0.73 | 0.99 | 0.46 | 0.99 | 0.46 | 0.60 |
| NR-Aromatase | 0.81 | 0.96 | 0.60 | 0.99 | 0.21 | 0.98 | 0.21 | 0.32 |
| NR-PPAR-gamma | 0.82 | 0.97 | 0.56 | 0.99 | 0.12 | 0.99 | 0.12 | 0.20 |
| SR-HSE | 0.77 | 0.95 | 0.56 | 0.99 | 0.13 | 0.97 | 0.13 | 0.22 |
| NR-AR-LBD | 0.85 | 0.98 | 0.77 | 0.99 | 0.54 | 0.99 | 0.54 | 0.66 |
| NR-ER | 0.72 | 0.89 | 0.60 | 0.98 | 0.21 | 0.93 | 0.21 | 0.32 |
| SR-ARE | 0.81 | 0.86 | 0.61 | 0.99 | 0.24 | 0.94 | 0.24 | 0.36 |
| SR-MMP | 0.88 | 0.89 | 0.67 | 0.98 | 0.36 | 0.95 | 0.36 | 0.49 |
| NR-AhR | 0.89 | 0.91 | 0.65 | 0.99 | 0.32 | 0.96 | 0.32 | 0.45 |
| NR-ER-LBD | 0.80 | 0.96 | 0.65 | 0.99 | 0.31 | 0.99 | 0.31 | 0.45 |
| SR-ATAD5 | 0.84 | 0.96 | 0.55 | 0.99 | 0.10 | 0.97 | 0.10 | 0.18 |
| SR-p53 | 0.83 | 0.95 | 0.56 | 0.99 | 0.13 | 0.99 | 0.13 | 0.22 |
| CT-TOX | 0.76 | 0.92 | 0.55 | 0.98 | 0.13 | 0.85 | 0.13 | 0.19 |
| Average | **0.81** | 0.94 | 0.62 | 0.99 | 0.25 | 0.96 | 0.25 | 0.36 |

Table G.3: Proportion of molecules being predicted to have a number of toxic endpoints. There are 200k molecules for each of the targets NSP9, RBD, and Main Protease. While there are only 680 molecules in the FDA database. Note that there are negligible amount of molecules or metabolites having more than 10 toxic endpoints and thus they are omitted here.

| Targets | 0 | 1 | 2 | 3 | 4 | 5 | 6 | 7 | 8 | 9 | 10 |
|---|---|---|---|---|---|---|---|---|---|---|---|
| Molecules |
| NSP9 | 0.502 | 0.223 | 0.121 | 0.076 | 0.042 | 0.021 | 0.01 | 0.004 | 0.001 | 0.0 | 0.0 |
| RBD | 0.505 | 0.223 | 0.121 | 0.075 | 0.041 | 0.021 | 0.009 | 0.004 | 0.001 | 0.0 | 0.0 |
| Main Protease | 0.507 | 0.221 | 0.123 | 0.073 | 0.04 | 0.021 | 0.009 | 0.004 | 0.001 | 0.0 | 0.0 |
| FDA | 0.569 | 0.157 | 0.1 | 0.056 | 0.038 | 0.034 | 0.022 | 0.013 | 0.006 | 0.001 | 0.001 |
| Metabolites |
| NSP9 | 0.642 | 0.191 | 0.075 | 0.038 | 0.023 | 0.014 | 0.009 | 0.005 | 0.002 | 0.0 | 0.0 |
| RBD | 0.63 | 0.206 | 0.085 | 0.04 | 0.019 | 0.011 | 0.005 | 0.002 | 0.001 | 0.0 | 0.0 |
| Main Protease | 0.652 | 0.192 | 0.071 | 0.038 | 0.021 | 0.011 | 0.008 | 0.004 | 0.002 | 0.0 | 0.0 |
| FDA | 0.656 | 0.138 | 0.068 | 0.045 | 0.037 | 0.025 | 0.013 | 0.01 | 0.004 | 0.002 | 0.001 |

# H   Docking Analysis

First, we removed chiral molecules from consideration for docking, as handling of chiral molecules in silico and in wet lab is tricky. We performed docking simulations using AutoDock Vina [43] with exhaustiveness=8 and a search space encompassing the entire protein target. We used the best result from 5 independent runs. Using a large set of approximately $875,000$ molecules (generated with only affinity constraints) for each target, we form clusters from the geometric centers of the top docking poses. We perform only 1 run of docking for these ligands. We use these cluster locations to approximate common binding sites for each target. We observe some correspondence with known binding pockets from literature — e.g., cluster 0 for $M^{pro}$ corresponds closely to the substrate-binding pocket [60] — as well as with pockets identified with PrankWeb [61, 62] (see Table H.3).

Our experiments also reveal that CogMol consistently yields higher percentage of low binding energy ($< -7$ kcal/mol) molecules compared to random ZINC ligands – this difference is 37%, 39% and 22% for RBD, MPro and NSP9 respectively.

Table H.1: Docking analysis of *screened* molecules: Size (%), average (± standard deviation) binding free energy, minimum binding free energy, the fraction of generated molecules with binding free energy $< -6$ kcal/mol for each cluster. In parentheses after target name: % of generated molecules that showed a binding free energy $< -6$ kcal/mol. Table 3 shows a condensed version of this table.

| Target | | Size (%) | Mean (kcal/mol) | Min (kcal/mol) | Low Energy (%) |
|---|---|---|---|---|---|
| NSP9 Dimer (87%) | cluster 0 | 67 | $-6.8 \pm 0.7$ | $-8.6$ | 88 |
| | cluster 1 | 22 | $-6.9 \pm 0.9$ | $-8.8$ | 85 |
| | cluster 2 | 9 | $-7.0 \pm 0.8$ | $-8.8$ | 80 |
| | cluster 3 | 3 | $-6.5 \pm 0.9$ | $-8.1$ | 73 |
| Main Protease (91%) | cluster 0 | 76 | $-7.2 \pm 0.8$ | $-9.5$ | 93 |
| | cluster 1 | 18 | $-6.9 \pm 0.8$ | $-9.2$ | 86 |
| | cluster 2 | 4 | $-7.0 \pm 0.5$ | $-7.8$ | 94 |
| | cluster 3 | 2 | $-7.0 \pm 1.0$ | $-8.4$ | 75 |
| | cluster 4 | 1 | $-6.8 \pm 1.3$ | $-8.2$ | 67 |
| RBD (95%) | cluster 0 | 30 | $-6.9 \pm 0.6$ | $-8.3$ | 93 |
| | cluster 1 | 36 | $-7.2 \pm 0.6$ | $-9.1$ | 97 |
| | cluster 2 | 18 | $-6.8 \pm 0.7$ | $-8.3$ | 84 |
| | cluster 3 | 12 | $-7.3 \pm 0.5$ | $-9.1$ | 100 |
| | cluster 4 | 3 | $-6.9 \pm 0.7$ | $-8.0$ | 86 |
| | cluster 5 | 1 | $-6.8 \pm 0.4$ | $-7.3$ | 100 |

Table H.2: Summary of docking for larger set of generated molecules: cluster size as a percentage of total molecules per target, average (± standard deviation) binding free energy, minimum binding free energy, the fraction of generated molecules with binding free energy $\leq -6$ kcal/mol. In parentheses after target name: 86%, 90%, and 84% of all generated molecules for the respective targets showed a binding free energy of $\leq -6$ kcal/mol. The molecules used were generated with just an affinity criterion and *not screened* for toxicity or retrosynthesis. These were used to fit the Gaussian Mixture Models and form clusters shown in Figure H.1.

| Target | | Size (%) | Mean (kcal/mol) | Min (kcal/mol) | Low Energy (%) |
|---|---|---|---|---|---|
| NSP9 Dimer (86%) | cluster 0 | 70 | $-6.9 \pm 0.8$ | $-10.7$ | 87 |
| | cluster 1 | 18 | $-6.9 \pm 0.9$ | $-12.0$ | 85 |
| | cluster 2 | 9 | $-7.0 \pm 0.8$ | $-10.5$ | 86 |
| | cluster 3 | 4 | $-6.6 \pm 0.8$ | $-9.8$ | 75 |
| Main Protease (90%) | cluster 0 | 58 | $-7.2 \pm 0.8$ | $-10.7$ | 92 |
| | cluster 1 | 26 | $-7.1 \pm 0.9$ | $-10.9$ | 88 |
| | cluster 2 | 11 | $-7.1 \pm 0.8$ | $-10.9$ | 91 |
| | cluster 3 | 2 | $-7.1 \pm 0.8$ | $-9.9$ | 89 |
| | cluster 4 | 2 | $-6.4 \pm 1.0$ | $-10.0$ | 63 |
| RBD (84%) | cluster 0 | 40 | $-6.7 \pm 0.8$ | $-10.6$ | 81 |
| | cluster 1 | 28 | $-7.0 \pm 0.8$ | $-12.4$ | 90 |
| | cluster 2 | 16 | $-6.6 \pm 0.7$ | $-9.6$ | 77 |
| | cluster 3 | 11 | $-7.1 \pm 0.7$ | $-10.7$ | 93 |
| | cluster 4 | 3 | $-6.7 \pm 0.8$ | $-9.8$ | 78 |
| | cluster 5 | 1 | $-6.9 \pm 0.7$ | $-10.5$ | 89 |

(a) NSP9

(b) M$^{\text{pro}}$

(c) RBD

Figure H.1: Binding cluster mean locations from blind docking results for NSP9 Replicase, Main Protease, and receptor-Binding Domain (RBD) of S protein of SARS-CoV-2. For each cluster, a large colored sphere is shown centered on the cluster mean. Note: these do not represent the exact extents of each cluster but simply serve to show the approximate locations. The indexes are used to refer to a specific cluster throughout this study.

Table H.3: Cluster mappings for identified binding pockets. Pockets are ordered according to descending score (a combination of predicted ligandability and conservation). Note: clusters encompass larger regions so multiple pockets may correspond to the same cluster.

| Target | | Cluster |
|---|---|---|
| NSP9 Dimer | pocket 0 | 2 |
| | pocket 1 | 0 |
| | pocket 2 | 1 |
| | pocket 3 | 1 |
| Main Protease (91%) | pocket 0 | 0 |
| | pocket 1 | 1 |
| | pocket 2 | 1 |
| | pocket 3 | 3 |
| RBD (95%) | pocket 0 | 5 |
| | pocket 1 | 4 |
| | pocket 2 | 0 |
| | pocket 3 | 1 |
| | pocket 4 | 2 |
| | pocket 5 | 3 |

(a) NSP9 Dimer (cluster 1)

(b) RBD (cluster 1)

(c) Main Protease (cluster 0)

Figure H.2: Predicted lowest energy binding mode for top generated molecule (in terms of docking binding free energy) with the corresponding target. Protein residues within 4 angstroms of the generated ligand are shown in cyan.

Figure H.3: Predicted lowest energy binding mode for PubChem Compound ID 76332092 with RBD (cluster 0).

# I Additional Details and Analysis of the Retrosynthesis

The generated molecules were tested for synthesizability, using a retrosynthetic algorithm [44]** based on the Molecular Transformer [45], trained on patent chemical reaction data. The performance of the retrosynthetic model depends on the type of chemistry needed to design a specific compound. Indeed, while chemical reactions from patents contain a wide variety of synthetic strategies, their variability is strongly biased by those reaction schemes that are mostly used in pharma and chemical industry. The need of academic type of reactions in the synthesis of a molecule may result in the entire design being not successful due to the lack of specific chemical knowledge. It is also important to remark that the retrosynthetic model [44] does not memorize data, but captures and learns chemical reaction patterns. The retrosynthesis is considered successful when, within the maximum allowed number of steps, the algorithm is able to reach commercially available materials All retrosynthesis have been executed with a maximum number of retrosynthetic steps equal to 6, a forward acceptance probability of 0.6, number of beams equal to 10, pruning steps equal to 2, no precursor price threshold and maximum execution time of 1 hour[††]. All commercially available products have been extracted from the emolecules database [63] restricting the selection to materials with a lead time of 4 weeks or less.

An important note on the FDA set considered in the analysis [52], it that among the 682 compounds only 489 molecules have been considered in the analysis after removal of entries mapping multiple chemicals, such as salts.

The reported number of steps (steps) was used to build correlation plots (see Figures I.1,I.2,I.3) among a large number of descriptors, including: a measure of the affinity (AFF), toxicity (TOX), selectivity (SEL), drug-likeness (QED), synthetic accessibility (SA), partition coefficient (LogP), the molecular weight (MolW), novelty (NOV). We report the correlation analysis for the $M^{pro}$ (see Figures I.1), NSP9 (see Figures I.2) and RBD target (see Figures I.3). As expected, we observe a positive correlation between SA and the number of steps. It is interesting to observe how the novelty (NOV) is positively correlated with the number of steps for both the Main Protease and the NSP9 Replicase, while for the RBD there is no evident correlation. This confirms that in the latter case, while the scaffolds are as novel as the $M^{pro}$ and the NSP9-related molecules, the type of precursors needed exhibit a greater variability of chemical complexity resulting in an uncorrelated number of steps needed to complete the synthesis.

Figure I.1: Main Protease correlation between properties and synthesis.

**The predictions have been performed using the python package `rxn4chemistry` (https://github.com/rxn4chemistry/rxn4chemistry)

††For a detailed description of all the parameters see [44] or `rxn4chemistry` docs: https://rxn4chemistry.github.io/rxn4chemistry/_modules/rxn4chemistry.html#rxn4chemistry.core.RXN4ChemistryWrapper.predict_automatic_retrosynthesis

Figure I.2: NSP9 Replicase correlation between molecule properties and synthesis.

Figure I.3: RBD of S Protein correlation between molecule properties and synthesis.

## J  Molecule Explorer

In order to allow domain experts to explore and screen the generated molecules for further analysis, we created an intuitive Molecule Explorer tool[‡‡]. The tools allows a user to filter molecules based on a variety of properties (QED, Target Affinity, Docking Energy, Synthetic Accessibility, Selectivity, Solubility and Novelty) and identify existing molecules in PubChem that are closest to the generated molecule. We present some screenshots of the molecule explorer in Figures J.1, J.2, and J.3. A screencast of the tool can be viewed at https://www.youtube.com/watch?v=cYb8_catBpI

The molecule explorer provides a user interface to prioritize and download the molecules using a variety of criteria. Along with the properties mentioned above, the tool also provides retrosynthesis routes for the generated molecules. This allows the user to select molecules based on the difficulty of synthesizing a molecule or based on the availability of components required to synthesize it. Another

---

[‡‡]http://ibm.biz/covid19-mol

criteria for prioritization could include the toxicity of the generated molecules to different end points, where certain endpoints are more important than others, depending on the application.

Figure J.1: Generated molecules for Main Protease with high affinity and low toxicity are displayed in a list.

Figure J.2: Generated molecules for NSP9 with low solubility and low toxicity, displayed in a plot of target affinity vs drug likeliness (QED).

# K  HDAC1: Molecule Generation for a Target with Low Coverage

As discussed in Section 5.3, we generated molecules that targets Human HDAC1. Figure K.1 shows the distribution of QED for molecules that are present in ZINC, BindingDB and the subset of molecules from BindingDB that binds to HDAC1. We can see that there are very few molecules with high QED that binds to HDAC1. Table K.1 shows that CogMol-generated molecules comprise a larger proportion of molecules satisfying high pIC50 and QED criteria, implying CogMol can discover novel and optimal molecules even in a low-data regime. Docking analysis of the generated molecules and comparison with training molecules are shown in Table K.2, showing the binding energy of the generated molecules are comparable with that of the training molecules.

Figure J.3: Generated molecules for Main Protease displayed in a plot format, here, the related molecules for GEN237 are shown via lines to those in the dataset. Sub-structure common to the molecules are highlighted on the left, with examples listed.

Figure K.1: Distribution of QED for molecules in ZINC, BindingDB and for molecules that bind with HDAC1 (pIC50>6)

Table K.1: Number of designed molecules with desired attributes compared to BindingDB ligands for HDAC1.

| Dataset | Total Molecules | QED>0.8 | pIC50>6 and QED >0.8 | pIC50 >7 and QED >0.8 |
|---|---|---|---|---|
| Train Set | 2253 | 43 (1.9%) | 9 (0.39%) | 1(0.04%) |
| Generated | 1388 | 188 (13.6%) | 89 (6.42%) | 32(2.3%) |

Table K.2: HDAC1 docking results: average ($\pm$ standard deviation) binding free energy, minimum binding free energy, and fraction of generated molecules with binding free energy $< -6$ kcal/mol for molecules from the train set (with QED > 0.7) and generated molecules.

| Dataset | Mean (kcal/mol) | Min (kcal/mol) | Low Energy (%) |
|---|---|---|---|
| Train Set | $-6.9 \pm 0.6$ | $-8.2$ | 92.5 |
| Generated | $-7.1 \pm 0.8$ | $-8.8$ | 89.9 |