[Reviews · NeurIPS 2020]

Review 1

Summary and Contributions: The authors propose a framework (CogMol) for drug-like small molecule design that considers target affinity, off-target selectivity, toxicity, binding complementarity, and synthetic accessibility. They claim that cogmol is capable of generating drug-like molecules with high affinity and selectivity for novel targets that have low ligand coverage. Utilizing a combination of generative models, the framework claims to generalize from 1D protein sequence information and generates molecules capable of binding to pockets of the 3D target structure with “favorable” binding free energy. Lastly, the authors claim to identify a set of novel and optimal drug-like molecules with high target affinity and selectivity for three novel COVID-19 targets and HDAC1.

Strengths: The use of VAEs for sampling from chemical space defined by the ZINC database can lead to sufficiently novel molecules determined by the objective function. The combination of VAE latent features and protein sequence embeddings can lead to accurate prediction of protein-ligand binding affinities. The focus on incorporating binding selectivity into the model building and evaluation process is important as promiscuous molecules can lead to toxicity and other clinical liabilities if not managed appropriately.

Weaknesses: The benchmarking of the molecular VAE model does not include a null model so as to assess its performance compared to random sampling of chemical space. The results show that generating high-affinity ligands is more challenging for NSP9 but the authors provide no reasoning or discussion as to why this may be. Could this be an artifact of the available training data in regards to its size and range of affinities? In the section on novelty, the authors mention using Tanimoto similarity between molecular fingerprints but do not delineate the specific algorithm and parameters used for fingerprint generation. Previous studies have demonstrated that the calculated similarities between molecules can vary significantly between fingerprinting methods. In addition, for binary topological fingerprinting methods, the number of bits defined to encode molecular topology and the radius of the atomic environments can lead to information degeneracy and differences in similarity calculations. Can the authors elaborate on the range of affinities in addition to the targets of all generated molecules that match exactly with an existing SMILES string in PubChem? What did the selectivity predictions say for this set of molecules? In the context of the docking calculations, can the authors describe what docking program, protocol, and protein/ligand setup protocol was utilized to make these predictions? Also, can the authors show evidence of what the expected docking score is for a random sample of ligands from the ZINC database to get a more quantitative understanding of whether the docking scores presented are significant? Docking scores do not account for the effects of dehydration and are often performed in the context of a rigid protein structure which can lead to poor correlation with experimental observables.

Correctness: The methodological claims appear correct however some claims in the conclusions of the study require more supporting and experimental data to provide sufficient evidence to support them (outlined above).

Clarity: The overall flow and prose of the paper is good.

Relation to Prior Work: Yes, the literature review is sufficient.

Reproducibility: No

Additional Feedback: I found the response from the authors to my review helpful, but they do not sufficiently address many of the reservations I have about this paper. Many of the claims in the paper are unsubstantiated and when asking for any experimental validation in my review for the claims in the paper the response was that extensive insilico validation was performed. This is not experimental validation and the authors must use language that is objective about this to help readers draw correct conclusions about the scope of this work. For example in line 335 "When applied to COVID-19 novel viral protein sequences, CogMol generated novel molecules that were able to bind favorably to the relevant druggable pockets of the target structure". In my view, this paper is dangerous to the community if published in its current form without reiterating that no experimental validation has been performed and updating statements regarding "binding" as "insilico binding". It would have been nice to see an updated manuscript with the revisions in place as well.


Review 2

Summary and Contributions: The authors propose a novel VAE-based framework called CogMol for proposing new putative drug molecules with binding affinity and selectivity to specified targets. They apply CogMol to the problem of designing molecules that bind to three key proteins found in SARS-CoV-19.

Strengths: 1) The authors are able to leverage a large database of existing compounds and their properties. 2) The authors consider multiple facets of their drugs, including the ease of synthesis and the binding affinity and selectivity. 3) The authors generate a database of over 3000 candidate molecules and share it in the public domain, along with a visualization tool to help facilitate their exploration.

Weaknesses: 1) Given that the targets are novel, the authors have no choice but to rely on an unvalidated folded structure, or use docking methods that only take their sequence into account. 2) The diverse set of methods used means that there is no underlying theoretical framework for the approach. 3) I would have liked to see more prioritization among the 3500 compounds that are generated by the approach; which are most likely to work? Easiest to synthesize? Most specific? These are all questions that I would have liked to see addressed more in detail.

Correctness: The methodology appears to be correct to the extent that I am able to understand it.

Clarity: The paper is fairly well written.

Relation to Prior Work: There is adequate reference and comparison to previous work.

Reproducibility: Yes

Additional Feedback:


Review 3

Summary and Contributions: The authors propose a drug candidate generation pipeline. It consists of various components, i.e., pre-training of molecular VAE generative model, controlled sampling scheme with deep learning attribute prediction models, and in silico screening with again deep learning prediction models. The authors provide extensive empirical evaluation results to demonstrate the effectiveness of the pipeline.

Strengths: (1) Novelty and significance Considering the pandemic situation, the paper tackles a critical problem. To the best of my knowledge, this would be one of the first works that proposed a deep generative model-based drug generation pipeline applicable to COVID-19. (2) Extensive empirical experiments Although it is impossible to tell whether the proposed method is truly effective without wet-lab evaluations, the authors have put great effort to computationally show its effectiveness. The experiment setup seems fair and the results are promising.

Weaknesses: While the proposed method is novel and the experiments provide impressive results, I am not quite sure that the authors provide enough details to reproduce the results. I am curious whether the authors plan to publicly release all the organized codes and datasets used in the experiments. Other than the reproducibility issue, I highly appreciate the authors’ effort in the extensive in silico evaluations and did not find other weaknesses.

Correctness: The claims and methods seem correct.

Clarity: The paper is generally well written and easy to understand. The contribution is clear, and the way the authors claim is well understandable and notations are explained well.

Relation to Prior Work: Yes. It provides a brief introduction of related paper and how it differs from them.

Reproducibility: No

Additional Feedback: No additional comments. ---- Post Author Feedback Comments --- I have read the Author Feedback and I'm happy that the authors have clarified some points. However, I agree with other reviewers that the authors must be more careful for overstatement. I do not think wet-lab experiments are required for the NeurIPS papers, but it should be stated more explicitly that only in silico evaluations are performed. The authors must let readers to know that proxy scores (e.g. docking scores) might not be significantly correlated with what happens in the real-world.


Review 4

Summary and Contributions: Their approach, CogMol, identifies new small molecule drugs that might provide a solution for coivd. They use a VAE using SMILES representations. They then train a protein-molecule binding affinity regressor, and an sampling approach using attributes. Their primary technical contribution is "O ur proposed methodology aims at computationally efficient targeted generation with multiple constraints from the latent space, requiring minimal model training and no supervision."(129) This allows for increased number of bindings which in tern should increased generalization.

Strengths: Rapid Prototyping is of obvious current concern, and they have positioned a solution for this problem. Their approach is well investigated for being an in silico process and builds on prior work to extend VAE's in application.

Weaknesses: This type of work will always suffer from assumptions inherent in using metrics like "The generated compounds are also comparable to FDA-approved drugs", but that shouldn't count against their analysis. Additionally, their ML contribution isn't radically different, but appears to indicate a logical next step. They counter this weakness by the completeness of their analysis.

Correctness: The approach appears grounded and well defined. They build on previous approaches and delineate the differences between their work and previous. They worked hard address concerns about the validity of the proposed results by checking for toxicity etc.

Clarity: The paper is well written but dense. There is a lot of biologic background that they work thru to cover the explanation of what they did.

Relation to Prior Work: This work provides good connection to prior work and delineates the difference between their results and others. It provides the necessary context to understand where their contribution is coming from. They provide ample opportunities to follow citations for determining the appropriate information to "recreate" previous work.

Reproducibility: Yes

Additional Feedback: Nice work. It is hard to balance ML clarity and Bio clarity. You've done a good job of providing enough information re the biological problem, potentially at the expense of ML clarity. I think you made the right decision in terms of balance.

[Author Response · NeurIPS 2020]

We thank all the reviewers for their thoughtful and constructive feedback. We are encouraged to hear that the reviewers
find the paper novel (**R2** , **R3** ), tackles a critical problem (**R3** ), well written (**R2** , **R3** , **R4** ), has extensive experiments
(**R3** ), is appropriately positioned with respect to prior work (**R2** , **R3** , **R4** ) and provides impressive results (**R3** ).
Moreover, **R4** acknowledges the completeness of our analysis and believes that we have worked hard to show the
validity of the proposed results. **R2** acknowledges that we incorporate multiple aspects of the generated molecules like
binding affinity, selectivity, toxicity, etc., in our pipeline. We answer specific questions posed by the reviewers below
and will incorporate all feedback in the revision. We apologize for the terseness of the replies due to space limitations.

**R1** : **Benchmarking of VAE with respect to a null model (random sampling of chemical space).** We use the
MOSES benchmarking framework for evaluating the molecular VAE model and perform extensive analysis of the
generated molecules in *Supp. Mat. Sec. B.* Comparison to other related molecular generative models (JT-VAE, AAE,
etc) is in *Supp. Mat. Tables B.1 and B.2*. Comparison of controlled vs random sampling is in *Table 1*. Our objective is
to show that controlled sampling on our VAE model can generate novel, diverse, targeted, drug-like molecules, when
trained on ZINC+BindingDB data. Molecules selected through random sampling of the chemical space will not have
these properties and hence are not included in the MOSES benchmarking framework as well.

**R1** : **Why generating high-affinity ligands is more challenging for NSP9?** In *Supp. Mat. Sec A*, we perform a
BLAST similarity search of MPro, RBD and NSP9 to the training data and show that NSP9 has the least sequence
similarity to the training data and is therefore more novel compared to other targets and hence more challenging.

**R1** : **Tanimoto similarity - specific algorithm and parameters used.** In *Supp. Mat. Sec E.*, we mention that we use
MACCS keys [58] for fingerprint generation. MACCS uses fixed 166-bit keys and has no additional parameters.

**R1** : **Details of docking program and protocol.** Details of the docking program (Autodock Vina) and the protocol are
explained in *Supp. Mat. Sec. H.*

**R1** : **Docking score for a random sample of ZINC ligands.** Our experiments reveal that CogMol consistently yields
higher percentage of low binding energy ($< -7$ kcal/mol) molecules compared to random ZINC ligands – this difference
is 37%, 39% and 22% for RBD, MPro and NSP9 respectively. We will add these results in Table 3 in final manuscript.

**R1** : **Affinities, targets, and selectivity of all generated molecules that match exactly with PubChem.** Avg.
Affinity/Selectivity/Fraction of matches for MPro (7.4/1.3/0.018), RBD (7.6/1.1/0.005), NSP9 (6.8/0.89/0.014).

**R1** : **Limitations of docking scores.** We agree with **R1** that the docking scores may provide limited information due to
simplification of various physical properties, such as the ignoring water and protein dynamics. However, recent work
(Gaillard et al., J. Chem. Inf. Model. 2018, 58, 8, 1697–1706) shows Autodock Vina scoring function is in the first
quarter (Vina) among all methods tested in CASF-2013 benchmark. Therefore, we use Vina scores as the first step
towards evaluating generated molecules in terms of target structure binding.

**R1** : **Some claims in the conclusions of the study require more supporting and experimental data to provide
sufficient evidence to support them.** We have performed extensive in silico experiments to validate the generated
molecules on multiple criteria (Parent Molecule and Metabolite Toxicity, Affinity based on an independent machine
learning model, Selectivity, Synthetic Feasibility, Target Structure Docking, Number of retrosynthetic steps, etc). See
*Section 5 and Supp. Materials*.

**R2** : **authors have no choice but to rely on an unvalidated folded structure, or use docking methods that only
take their sequence into account.** Docking was done using x-ray crystal structures of target proteins available in
Protein Data Bank, whereas the generative model relied on protein sequence information only.

**R2** : **The diverse set of methods used means that there is no underlying theoretical framework for the approach.**
Our work is the first deep generative approach that generates novel, specific, and selective drug-like small molecules for
an unseen target protein sequence without requiring protein-specific model retraining. The diverse set of screening
methods provide a comprehensive analysis of generated molecules to validate the generated molecules and show the
efficacy of the generative framework.

**R2** : **Prioritization among the 3500 compounds that are generated by the approach.** Great suggestion! We will
provide a discussion around different prioritization schemes. For example, one scheme will be based on number of
synthesis steps, and the cost and availability of ingredients.

**R3** : **Reproducibility.** All training datasets are publicly available. Also, the generated molecules and their properties
are available for download using our molecule explorer tool. We did not provide a URL of the tool to keep our
submission anonymous during review, but will add it upon acceptance. Some of the molecular evaluation components
are already publicly available (*e.g.* rxn4chemistry package , Autodock Vina) We have also provided extensive details of
the architecture and parameters of our models in the *Supp. Mat.* and will add any specific details we may have missed.
We are also working on releasing the source code of some other components upon acceptance.

[Meta-Review · NeurIPS 2020]

This paper proposes a framework, called CogMol, to design a drug-like small molecule for specific targets, which was applied to the problem of designing molecules that bind to three proteins found in SARS-CoV-19. Reviewers raised various concerns and questions and author response largely resolved major criticisms. Overall, based on the technical novelty, experiments, and clarity in writing, this paper passes the bar of acceptance to NeurIPS as a technical paper. However, multiple reviewers expressed a concern about the possibility that readers over-interpret the results in the context of the current pandemic situation, because wet-lab validation experiments have not been performed, (which would be out of scope and not necessary for a ML conference paper.) Thus, it is strongly recommended that the authors revise the manuscript to explicitly state that no experimental validation has been performed and only in-silico binding conclusions can be drawn.